# Unsupervised Learning of Disentangled and Interpretable Representations from Sequential Data

**Wei-Ning Hsu, Yu Zhang, and James Glass**
Computer Science and Artificial Intelligence Laboratory
Massachusetts Institute of Technology
Cambridge, MA 02139, USA
{wnhsu,yzhang87,glass}@csail.mit.edu

## Abstract

We present a factorized hierarchical variational autoencoder, which learns disentangled and interpretable representations from sequential data without supervision. Specifically, we exploit the multi-scale nature of information in sequential data by formulating it explicitly within a factorized hierarchical graphical model that imposes sequence-dependent priors and sequence-independent priors to different sets of latent variables. The model is evaluated on two speech corpora to demonstrate, qualitatively, its ability to transform speakers or linguistic content by manipulating different sets of latent variables; and quantitatively, its ability to outperform an i-vector baseline for speaker verification and reduce the word error rate by as much as 35% in mismatched train/test scenarios for automatic speech recognition tasks.

## 1 Introduction

Unsupervised learning is a powerful methodology that can leverage vast quantities of unannotated data in order to learn useful representations that can be incorporated into subsequent applications in either supervised or unsupervised fashions. One of the principle approaches to unsupervised learning is probabilistic generative modeling. Recently, there has been significant interest in three classes of deep probabilistic generative models: 1) Variational Autoencoders (VAEs) [23, 34, 22], 2) Generative Adversarial Networks (GANs) [11], and 3) auto-regressive models [30, 39]; more recently, there are also studies combining multiple classes of models [6, 27, 26]. While GANs bypass any inference of latent variables, and auto-regressive models abstain from using latent variables, VAEs jointly learn an inference model and a generative model, allowing them to infer latent variables from observed data.

Despite successes with VAEs, understanding the underlying factors that latent variables associate with is a major challenge. Some research focuses on the supervised or semi-supervised setting using VAEs [21, 17]. There is also research attempting to develop weakly supervised or unsupervised methods to learn disentangled representations, such as DC-IGN [25], InfoGAN [1], and $\beta$-VAE [13]. There is yet another line of research analyzing the latent variables with labeled data after the model is trained [33, 15]. While there has been much research investigating static data, such as the aforementioned ones, there is relatively little research on learning from sequential data [8, 3, 2, 9, 7, 18, 36]. Moreover, to the best of our knowledge, there has not been any attempt to learn disentangled and interpretable representations without supervision from sequential data. The information encoded in sequential data, such as speech, video, and text, is naturally multi-scaled; in speech for example, information about the channel, speaker, and linguistic content is encoded in the statistics at the session, utterance, and segment levels, respectively. By leveraging this source of constraint, we can learn disentangled and interpretable factors in an unsupervised manner.

In this paper, we propose a novel factorized hierarchical variational autoencoder, which learns disentangled and interpretable latent representations from sequential data without supervision by

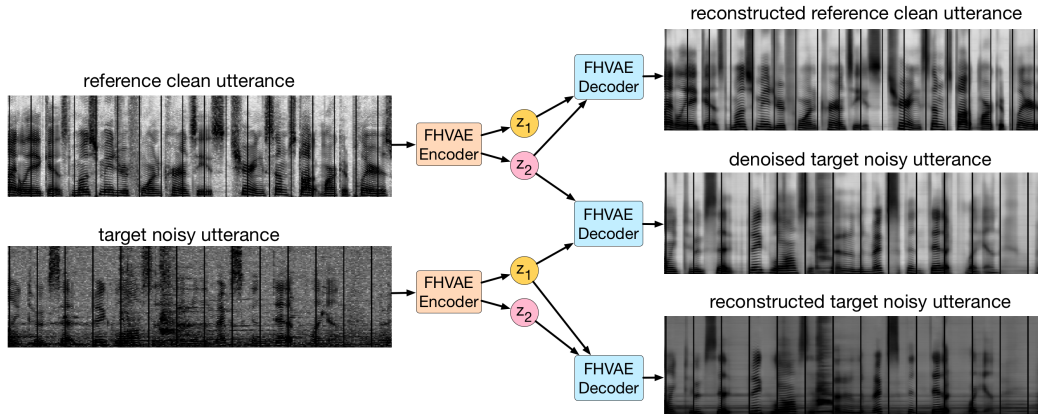

Figure 1: FHVAE ($\alpha = 0$) decoding results of three combinations of *latent segment variables* $z_1$ and *latent sequence variables* $z_2$ from two utterances in Aurora-4: a clean one (top-left) and a noisy one (bottom-left). FHVAEs learn to encode local attributes, such as linguistic content, into $z_1$, and encode global attributes, such as noise level, into $z_2$. Therefore, by replacing $z_2$ of a noisy utterance with $z_2$ of a clean utterance, an FHVAE decodes a denoised utterance (middle-right) that preserves the linguistic content. Reconstruction results of the clean and noisy utterances are also shown on the right. Audio samples are available at `https://youtu.be/naJZITvCfI4`.

explicitly modeling the multi-scaled information with a factorized hierarchical graphical model. The inference model is designed such that the model can be optimized at the segment level, instead of at the sequence level, which may cause scalability issues when sequences become too long. A sequence-to-sequence neural network architecture is applied to better capture temporal relationships. We evaluate the proposed model on two speech datasets. Qualitatively, the model demonstrates an ability to factorize sequence-level and segment-level attributes into different sets of latent variables. Quantitatively, the model achieves 2.38% and 1.34% equal error rate on unsupervised and supervised speaker verification tasks respectively, which outperforms an i-vector baseline. On speech recognition tasks, it reduces the word error rate in mismatched train/test scenarios by up to 35%.

The rest of the paper is organized as follows. In Section 2, we introduce our proposed model, and describe the neural network architecture in Section 3. Experimental results are reported in Section 4. We discuss related work in Section 5, and conclude our work as well as discuss future research plans in Section 6. We have released the code for the model described in this paper.[1]

## 2 Factorized Hierarchical Variational Autoencoder

Generation of sequential data, such as speech, often involves multiple independent factors operating at different time scales. For instance, the speaker identity affects fundamental frequency (F0) and volume at the sequence level, while phonetic content only affects spectral contour and durations of formants at the segmental level. This multi-scale behavior results in the fact that some attributes, such as F0 and volume, tend to have a smaller amount of variation within an utterance, compared to between utterances; while other attributes, such as phonetic content, tend to have a similar amount of variation within and between utterances.

We refer to the first type of attributes as *sequence-level attributes*, and the other as *segment-level attributes*. In this work, we achieve disentanglement and interpretability by encoding the two types of attributes into *latent sequence variables* and *latent segment variables* respectively, where the former is regularized by an sequence-dependent prior and the latter by an sequence-independent prior.

We now formulate a generative process for speech and propose our Factorized Hierarchical Variational Autoencoder (FHVAE). Consider some dataset $\mathcal{D} = \{\boldsymbol{X}^{(i)}\}_{i=1}^{M}$ consisting of $M$ i.i.d. sequences, where $\boldsymbol{X}^{(i)} = \{\boldsymbol{x}^{(i,n)}\}_{n=1}^{N^{(i)}}$ is a sequence of $N^{(i)}$ observed variables. $N^{(i)}$ is referred to as the

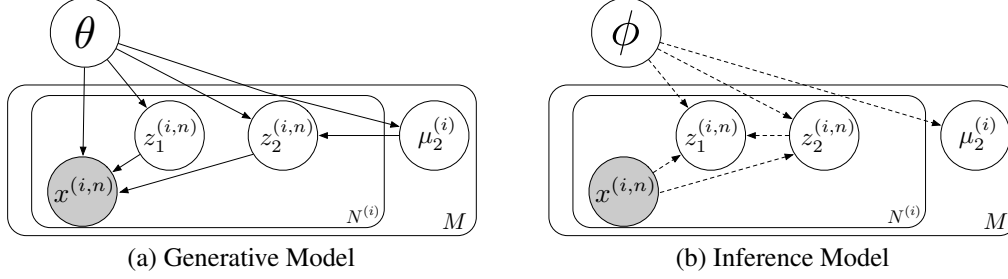

| (a) Generative Model | (b) Inference Model |

Figure 2: Graphical illustration of the proposed generative model and inference model. Grey nodes denote the observed variables, and white nodes are the hidden variables.

*number of segments* for the $i$-th sequence, and $\boldsymbol{x}^{(i,n)}$ is referred to as the $n$-th *segment* of the $i$-th sequence. Note that here a "segment" refers to a variable of smaller temporal scale compared to the "sequence", which is in fact a sub-sequence. We will drop the index $i$ whenever it is clear that we are referring to terms associated with a single sequence. We assume that each sequence $\boldsymbol{X}$ is generated from some random process involving the latent variables $\boldsymbol{Z}_1 = \{\boldsymbol{z}_1^{(n)}\}_{n=1}^N$, $\boldsymbol{Z}_2 = \{\boldsymbol{z}_2^{(n)}\}_{n=1}^N$, and $\boldsymbol{\mu}_2$. The following generation process as illustrated in Figure 2(a) is considered: (1) a *s-vector* $\boldsymbol{\mu}_2$ is drawn from a prior distribution $p_\theta(\boldsymbol{\mu}_2)$; (2) $N$ i.i.d. *latent sequence variables* $\{\boldsymbol{z}_2^{(n)}\}_{n=1}^N$ and *latent segment variables* $\{\boldsymbol{z}_1^{(n)}\}_{n=1}^N$ are drawn from a sequence-dependent prior distribution $p_\theta(\boldsymbol{z}_2|\boldsymbol{\mu}_2)$ and a sequence-independent prior distribution $p_\theta(\boldsymbol{z}_1)$ respectively; (3) $N$ i.i.d. observed variables $\{\boldsymbol{x}^{(n)}\}_{n=1}^N$ are drawn from a conditional distribution $p_\theta(\boldsymbol{x}|\boldsymbol{z}_1, \boldsymbol{z}_2)$. The joint probability for a sequence is formulated in Eq. 1:

$$p_\theta(\boldsymbol{X}, \boldsymbol{Z}_1, \boldsymbol{Z}_2, \boldsymbol{\mu}_2) = p_\theta(\boldsymbol{\mu}_2) \prod_{n=1}^N p_\theta(\boldsymbol{x}^{(n)}|\boldsymbol{z}_1^{(n)}, \boldsymbol{z}_2^{(n)})p_\theta(\boldsymbol{z}_1^{(n)})p_\theta(\boldsymbol{z}_2^{(n)}|\boldsymbol{\mu}_2). \quad (1)$$

Specifically, we formulate each of the RHS term as follows:

$$p_\theta(\boldsymbol{x}|\boldsymbol{z}_1, \boldsymbol{z}_2) = \mathcal{N}(\boldsymbol{x}|f_{\mu_x}(\boldsymbol{z}_1, \boldsymbol{z}_2), diag(f_{\sigma_x^2}(\boldsymbol{z}_1, \boldsymbol{z}_2)))$$

$$p_\theta(\boldsymbol{z}_1) = \mathcal{N}(\boldsymbol{z}_1|\boldsymbol{0}, \sigma_{\boldsymbol{z}_1}^2\boldsymbol{I}), \quad p_\theta(\boldsymbol{z}_2|\boldsymbol{\mu}_2) = \mathcal{N}(\boldsymbol{z}_2|\boldsymbol{\mu}_2, \sigma_{\boldsymbol{z}_2}^2\boldsymbol{I}), \quad p_\theta(\boldsymbol{\mu}_2) = \mathcal{N}(\boldsymbol{\mu}_2|\boldsymbol{0}, \sigma_{\boldsymbol{\mu}_2}^2\boldsymbol{I}),$$

where the priors over the s-vectors $\boldsymbol{\mu}_2$ and the latent segment variables $\boldsymbol{z}_1$ are centered isotropic multivariate Gaussian distributions. The prior over the latent sequence variable $\boldsymbol{z}_2$ conditioned on $\boldsymbol{\mu}_2$ is an isotropic multivariate Gaussian centered at $\boldsymbol{\mu}_2$. The conditional distribution of the observed variable $\boldsymbol{x}$ is the multivariate Gaussian with a diagonal covariance matrix, whose mean and diagonal variance are parameterized by neural networks $f_{\mu_x}(\cdot, \cdot)$ and $f_{\sigma_x^2}(\cdot, \cdot)$ with input $\boldsymbol{z}_1$ and $\boldsymbol{z}_2$. We use $\theta$ to denote the set of parameters in the generative model.

This generative model is factorized in a way such that the latent sequence variables $\boldsymbol{z}_2$ within a sequence are forced to be close to $\boldsymbol{\mu}_2$ as well as to each other in Euclidean distance, and therefore are encouraged to encode sequence-level attributes that may have larger variance across sequences, but smaller variance within sequences. The constraint to the latent segment variables $\boldsymbol{z}_1$ is imposed globally, and therefore encourages encoding of residual attributes whose variation is not distinguishable inter and intra sequences.

In the variational autoencoder framework, since the exact posterior inference is intractable, an inference model, $q_\phi(\boldsymbol{Z}_1^{(i)}, \boldsymbol{Z}_2^{(i)}, \boldsymbol{\mu}_2^{(i)}|\boldsymbol{X}^{(i)})$, that approximates the true posterior, $p_\theta(\boldsymbol{Z}_1^{(i)}, \boldsymbol{Z}_2^{(i)}, \boldsymbol{\mu}_2^{(i)}|\boldsymbol{X}^{(i)})$, for variational inference [19] is introduced. We consider the following inference model as Figure 2(b):

$$q_\phi(\boldsymbol{Z}_1^{(i)}, \boldsymbol{Z}_2^{(i)}, \boldsymbol{\mu}_2^{(i)}|\boldsymbol{X}^{(i)}) = q_\phi(\boldsymbol{\mu}_2^{(i)}) \prod_{n=1}^{N^{(i)}} q_\phi(\boldsymbol{z}_1^{(i,n)}|\boldsymbol{x}^{(i,n)}, \boldsymbol{z}_2^{(i,n)})q_\phi(\boldsymbol{z}_2^{(i,n)}|\boldsymbol{x}^{(i,n)})$$

$$q_\phi(\boldsymbol{\mu}_2^{(i)}) = \mathcal{N}(\boldsymbol{\mu}_2^{(i)}|g_{\mu_{\boldsymbol{\mu}_2}}(i), \sigma_{\tilde{\boldsymbol{\mu}}_2}^2\boldsymbol{I}), \quad q_\phi(\boldsymbol{z}_2|\boldsymbol{x}) = \mathcal{N}(\boldsymbol{z}_2|g_{\mu_{\boldsymbol{z}_2}}(\boldsymbol{x}), diag(g_{\sigma_{\boldsymbol{z}_2}^2}(\boldsymbol{x})))$$

$$q_\phi(\boldsymbol{z}_1|\boldsymbol{x}, \boldsymbol{z}_2) = \mathcal{N}(\boldsymbol{z}_1|g_{\mu_{\boldsymbol{z}_1}}(\boldsymbol{x}, \boldsymbol{z}_2), diag(g_{\sigma_{\boldsymbol{z}_1}^2}(\boldsymbol{x}, \boldsymbol{z}_2))),$$

where the posteriors over $\boldsymbol{\mu}_2$, $\boldsymbol{z}_1$, and $\boldsymbol{z}_2$ are all multivariate diagonal Gaussian distributions. Note that the mean of the posterior distribution of $\boldsymbol{\mu}_2$ is not directly inferred from $\boldsymbol{X}$, but instead is regarded as part of inference model parameters, with one for each utterance, which would be optimized during training. Therefore, $g_{\mu_{\mu_2}}(\cdot)$ can be seen as a lookup table, and we use $\tilde{\boldsymbol{\mu}}_2^{(i)} = g_{\mu_{\mu_2}}(i)$ to denote the posterior mean of $\boldsymbol{\mu}_2$ for the $i$-th sequence; we fix the posterior covariance matrix of $\boldsymbol{\mu}_2$ for all sequences. Similar to the generative model, $g_{\mu_{z_2}}(\cdot)$, $g_{\sigma_{z_2}^2}(\cdot)$, $g_{\mu_{z_1}}(\cdot,\cdot)$, and $g_{\sigma_{z_1}^2}(\cdot,\cdot)$ are also neural networks whose parameters along with $g_{\mu_{\mu_2}}(\cdot)$ are denoted collectively by $\phi$. The variational lower bound for this inference model on the marginal likelihood of a sequence $\boldsymbol{X}$ is derived as follows:

$$\mathcal{L}(\theta, \phi; \boldsymbol{X}) = \sum_{n=1}^{N} \mathcal{L}(\theta, \phi; \boldsymbol{x}^{(n)} | \tilde{\boldsymbol{\mu}}_2) + \log p_\theta(\tilde{\boldsymbol{\mu}}_2) + const$$

$$\begin{aligned}
\mathcal{L}(\theta, \phi; \boldsymbol{x}^{(n)} | \tilde{\boldsymbol{\mu}}_2) =& \mathbb{E}_{q_\phi(\boldsymbol{z}_1^{(n)}, \boldsymbol{z}_2^{(n)} | \boldsymbol{x}^{(n)})} \big[ \log p_\theta(\boldsymbol{x}^{(n)} | \boldsymbol{z}_1^{(n)}, \boldsymbol{z}_2^{(n)}) \big] \\
&- \mathbb{E}_{q_\phi(\boldsymbol{z}_2^{(n)} | \boldsymbol{x}^{(n)})} \big[ D_{KL}(q_\phi(\boldsymbol{z}_1^{(n)} | \boldsymbol{x}^{(n)}, \boldsymbol{z}_2^{(n)}) || p_\theta(\boldsymbol{z}_1^{(n)})) \big] \\
&- D_{KL}(q_\phi(\boldsymbol{z}_2^{(n)} | \boldsymbol{x}^{(n)}) || p_\theta(\boldsymbol{z}_2^{(n)} | \tilde{\boldsymbol{\mu}}_2)).
\end{aligned}$$

The detailed derivation can be found in Appendix A. Because the approximated posterior of $\boldsymbol{\mu}_2$ does not depend on the sequence $\boldsymbol{X}$, the *sequence variational lower bound* $\mathcal{L}(\theta, \phi; \boldsymbol{X})$ can be decomposed into the sum of $\mathcal{L}(\theta, \phi; \boldsymbol{x}^{(n)} | \tilde{\boldsymbol{\mu}}_2)$, the *conditional segment variational lower bounds*, over segments, plus the log prior probability of $\tilde{\boldsymbol{\mu}}_2$ and a constant. Therefore, instead of sampling a batch at the sequence level to maximize the sequence variational lower bound, we can sample a batch at the segment level to maximize the *segment variational lower bound*:

$$\mathcal{L}(\theta, \phi; \boldsymbol{x}^{(n)}) = \mathcal{L}(\theta, \phi; \boldsymbol{x}^{(n)} | \tilde{\boldsymbol{\mu}}_2) + \frac{1}{N} \log p_\theta(\tilde{\boldsymbol{\mu}}_2) + const. \tag{2}$$

This approach provides better scalability when the sequences are extremely long, such that computing an entire sequence for a batched update is too computationally expensive.

In this paper we only introduce two scales of attributes; however, one can easily extend this model to more scales by simply introducing $\boldsymbol{\mu}_k$ for $k = 2, 3, \cdots$ [2] that constrains the prior distribution of latent variables at more scales, such as having session-dependent prior or dataset-dependent prior.

## 2.1 Discriminative Objective

The idea of having sequence-specific priors for each sequence is to encourage the model to encode the sequence-level attributes and the segment-level attributes into different sets of latent variables. However, when $\boldsymbol{\mu}_2 = 0$ for all sequences, the prior probability of the s-vector is maximized, and the KL-divergence of the inferred posterior of $\boldsymbol{z}_2$ is measured from the same conditional prior for all sequences. This would result in trivial s-vectors $\boldsymbol{\mu}_2$, and therefore $\boldsymbol{z}_1$ and $\boldsymbol{z}_2$ would not be factorized to encode sequence and segment attributes respectively.

To encourage $\boldsymbol{z}_2$ to encode sequence-level attributes, we use $\boldsymbol{z}_2^{(i,n)}$, which is inferred from $\boldsymbol{x}^{(i,n)}$, to infer the sequence index $i$ of $\boldsymbol{x}^{(i,n)}$. We formulate the discriminative objective as:

$$\log p(i | \boldsymbol{z}_2^{(i,n)}) = \log p(\boldsymbol{z}_2^{(i,n)} | i) - \log \sum_{j=1}^{M} p(\boldsymbol{z}_2^{(i,n)} | j) \quad (p(i) \text{ is assumed uniform})$$

$$:= \log p_\theta(\boldsymbol{z}_2^{(i,n)} | \tilde{\boldsymbol{\mu}}_2^{(i)}) - \log \Big( \sum_{j=1}^{M} p_\theta(\boldsymbol{z}_2^{(i,n)} | \tilde{\boldsymbol{\mu}}_2^{(j)}) \Big),$$

Combining the discriminative objective using a weighting parameter $\alpha$ with the segment variational lower bound, the objective function to maximize then becomes:

$$\mathcal{L}^{dis}(\theta, \phi; \boldsymbol{x}^{(i,n)}) = \mathcal{L}(\theta, \phi; \boldsymbol{x}^{(i,n)}) + \alpha \log p(i | \boldsymbol{z}_2^{(i,n)}), \tag{3}$$

which we refer to as the *discriminative segment variational lower bound*.

## 2.2 Inferring S-Vectors During Testing

During testing, we may want to use the s-vector $\boldsymbol{\mu}_2$ of an unseen sequence $\tilde{\boldsymbol{X}} = \{\tilde{\boldsymbol{x}}^{(n)}\}_{n=1}^{\tilde{N}}$ as the sequence-level attribute representation for tasks such as speaker verification. Since the exact maximum a posterior estimation of $\boldsymbol{\mu}_2$ is intractable, we approximate the estimation using the conditional segment variational lower bound as follows:

$$\boldsymbol{\mu}_2^* = \underset{\boldsymbol{\mu}_2}{\operatorname{argmax}} \log p_\theta(\boldsymbol{\mu}_2 | \tilde{\boldsymbol{X}}) = \underset{\boldsymbol{\mu}_2}{\operatorname{argmax}} \log p_\theta(\tilde{\boldsymbol{X}}, \boldsymbol{\mu}_2)$$

$$= \underset{\boldsymbol{\mu}_2}{\operatorname{argmax}} \big( \sum_{n=1}^{\tilde{N}} \log p_\theta(\tilde{\boldsymbol{x}}^{(n)} | \boldsymbol{\mu}_2) \big) + \log p_\theta(\boldsymbol{\mu}_2)$$

$$\approx \underset{\boldsymbol{\mu}_2}{\operatorname{argmax}} \sum_{n=1}^{\tilde{N}} \mathcal{L}(\theta, \phi; \tilde{\boldsymbol{x}}^{(n)} | \boldsymbol{\mu}_2) + \log p_\theta(\boldsymbol{\mu}_2). \quad (4)$$

The closed form solution of $\boldsymbol{\mu}_2^*$ can be derived by differentiating Eq. 4 w.r.t. $\boldsymbol{\mu}_2$ (see Appendix B):

$$\boldsymbol{\mu}_2^* = \frac{\sum_{n=1}^{\tilde{N}} g_{\mu_{\boldsymbol{z}_2}}(\tilde{\boldsymbol{x}}^{(n)})}{\tilde{N} + \sigma_{\boldsymbol{z}_2}^2 / \sigma_{\boldsymbol{\mu}_2}^2}. \quad (5)$$

## 3 Sequence-to-Sequence Autoencoder Model Architecture

In this section, we introduce the detailed neural network architectures for our proposed FHVAE. Let a segment $\boldsymbol{x} = x_{1:T}$ be a sub-sequence of $\boldsymbol{X}$ that contains $T$ time steps, and $x_t$ denotes the $t$-th time step of $\boldsymbol{x}$. We use recurrent network architectures for encoders that capture the temporal relationship among time steps, and generate a summarized fixed-dimension vector after consuming an entire sub-sequence. Likewise, we adopt a recurrent network architecture that generates a frame step by step conditioned on the latent variables $\boldsymbol{z}_1$ and $\boldsymbol{z}_2$. The complete network can be seen as a stochastic sequence-to-sequence autoencoder that encodes $x_{1:T}$ stochastically into $\boldsymbol{z}_1, \boldsymbol{z}_2$, and stochastically decodes from them back to $x_{1:T}$.

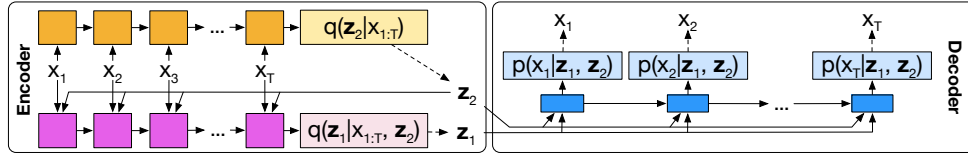

Figure 3: Sequence-to-sequence factorized hierarchical variational autoencoder. Dashed lines indicate the sampling process using the reparameterization trick [23]. The encoders for $\boldsymbol{z}_1$ and $\boldsymbol{z}_2$ are pink and amber, respectively, while the decoder for $\boldsymbol{x}$ is blue. Darker colors denote the recurrent neural networks, while lighter colors denote the fully-connected layers predicting the mean and log variance.

Figure 3 shows our proposed Seq2Seq-FHVAE architecture.[3] Here we show the detailed formulation:

$$(\boldsymbol{h}_{\boldsymbol{z}_2,t}, \boldsymbol{c}_{\boldsymbol{z}_2,t}) = \text{LSTM}(x_{t-1}, \boldsymbol{h}_{\boldsymbol{z}_2,t-1}, \boldsymbol{c}_{\boldsymbol{z}_2,t-1}; \phi_{\text{LSTM},\boldsymbol{z}_2})$$

$$q_\phi(\boldsymbol{z}_2 | x_{1:T}) = \mathcal{N}(\boldsymbol{z}_2 | \text{MLP}(\boldsymbol{h}_{\boldsymbol{z}_2,T}; \phi_{\text{MLP}_{\boldsymbol{\mu}},\boldsymbol{z}_2}), \text{diag}(\exp(\text{MLP}(\boldsymbol{h}_{\boldsymbol{z}_2,T}; \phi_{\text{MLP}_{\boldsymbol{\sigma}^2},\boldsymbol{z}_2}))))$$

$$(\boldsymbol{h}_{\boldsymbol{z}_1,t}, \boldsymbol{c}_{\boldsymbol{z}_1,t}) = \text{LSTM}([x_{t-1}; \boldsymbol{z}_2], \boldsymbol{h}_{\boldsymbol{z}_1,t-1}, \boldsymbol{c}_{\boldsymbol{z}_1,t-1}; \phi_{\boldsymbol{z}_1})$$

$$q_\phi(\boldsymbol{z}_1 | x_{1:T}, \boldsymbol{z}_2) = \mathcal{N}(\boldsymbol{z}_1 | \text{MLP}(\boldsymbol{h}_{\boldsymbol{z}_1,T}; \phi_{\text{MLP}_{\boldsymbol{\mu}},\boldsymbol{z}_1}), \text{diag}(\exp(\text{MLP}(\boldsymbol{h}_{\boldsymbol{z}_1,T}; \phi_{\text{MLP}_{\boldsymbol{\sigma}^2},\boldsymbol{z}_1}))))$$

$$(\boldsymbol{h}_{\boldsymbol{x},t}, \boldsymbol{c}_{\boldsymbol{x},t}) = \text{LSTM}([\boldsymbol{z}_1; \boldsymbol{z}_2], \boldsymbol{h}_{\boldsymbol{x},t-1}, \boldsymbol{c}_{\boldsymbol{x},t-1}; \phi_{\boldsymbol{x}})$$

$$p_\theta(x_t | \boldsymbol{z}_1, \boldsymbol{z}_2) = \mathcal{N}(\boldsymbol{x}_t | \text{MLP}(\boldsymbol{h}_{\boldsymbol{x},t}; \phi_{\text{MLP}_{\boldsymbol{\mu}},\boldsymbol{x}}), \text{diag}(\exp(\text{MLP}(\boldsymbol{h}_{\boldsymbol{x},t}; \phi_{\text{MLP}_{\boldsymbol{\sigma}^2},\boldsymbol{x}})))),$$

where LSTM refers to a long short-term memory recurrent neural network [14], and MLP refers to a multi-layer perceptron, $\phi_*$ are the related weight matrices. None of the neural network parameters are shared. We refer to this model as Seq2Seq-FHVAE. Log-likelihood and qualitative comparison with alternative architectures can be found in Appendix D.

# 4 Experiments

We use speech, which inherently contains information at multiple scales, such as channel, speaker, and linguistic content to test our model. Learning to disentangle the mixed information from the surface representation is essential for a wide variety of speech applications: for example, noise robust speech recognition [41, 38, 37, 16], speaker verification [5], and voice conversion [40, 29, 24].

The following two corpora are used for our experiments: (1) **TIMIT** [10], which contains broadband 16kHz recordings of phonetically-balanced read speech. A total of 6300 utterances (5.4 hours) are presented with 10 sentences from each of 630 speakers, of which approximately 70% are male and 30% are female. (2) **Aurora-4** [32], a broadband corpus designed for noisy speech recognition tasks based on the Wall Street Journal corpus (WSJ0) [31]. Two microphone types, CLEAN/CHANNEL are included, and six noise types are artificially added to both microphone types, which results in four conditions: CLEAN, CHANNEL, NOISY, and CHANNEL+NOISY. Two 14 hour training sets are used, where one is clean and the other is a mix of all four conditions. The same noise types and microphones are used to generate the development and test sets, which both consist of 330 utterances from all four conditions, resulting in 4,620 utterances in total for each set.

All speech is represented as a sequence of 80 dimensional Mel-scale filter bank (FBank) features or 200 dimensional log-magnitude spectrum (only for audio reconstruction), computed every 10ms. Mel-scale features are a popular auditory approximation for many speech applications [28]. We consider a sample $x$ to be a 200ms sub-sequence, which is on the order of the length of a syllable, and implies $T = 20$ for each $x$. For the Seq2Seq-FHVAE model, all the LSTM and MLP networks are one-layered, and Adam [20] is used for optimization. More details of the model architecture and training procedure can be found in Appendix C.

## 4.1 Qualitative Evaluation of the Disentangled Latent Variables

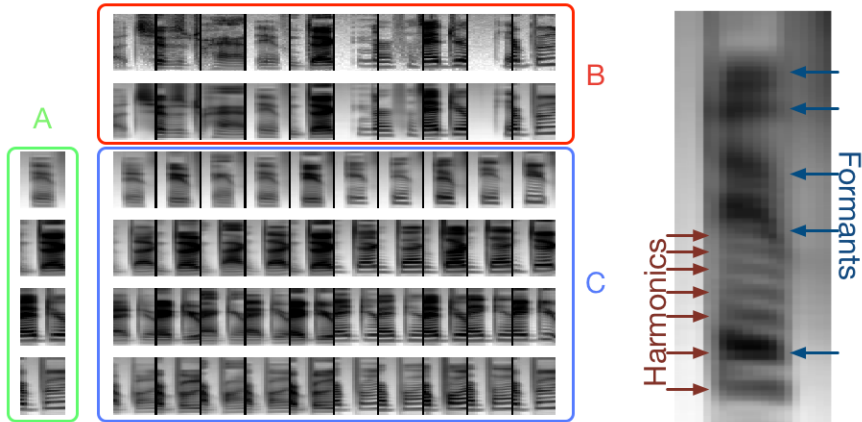

Figure 4: (left) Examples generated by varying different latent variables. (right) An illustration of harmonics and formants in filter bank images. The green block 'A' contains four reconstructed examples. The red block 'B' contains ten original sequences on the first row with the corresponding reconstructed examples on the second row. The entry on the $i$-th row and the $j$-th column in the blue block 'C' is the reconstructed example using the latent segment variable $z_1$ of the $i$-th row from block 'A' and the latent sequence variable $z_2$ of the $j$-th column from block 'B'.

To qualitatively study the factorization of information between the latent segment variable $z_1$ and the latent sequence variable $z_2$, we generate examples $x$ by varying each of them respectively. Figure 4 shows 40 examples in block 'C' of all the combinations of the 4 latent segment variables extracted from block 'A' and the 10 latent sequence variables extracted from block 'B.' The top two examples from block 'A' and the five leftmost examples from block 'B' are from male speakers, while the rest are from female speakers, which show higher fundamental frequencies and harmonics.[4]

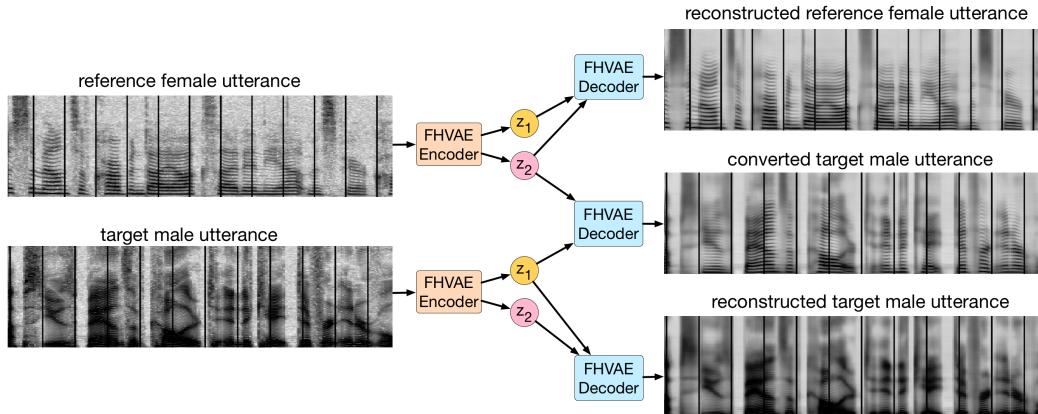

Figure 5: FHVAE ($\alpha = 0$) decoding results of three combinations of *latent segment variables* $z_1$ and *latent sequence variables* $z_2$ from one male-speaker utterance (top-left) and one female-speaker utterance (bottom-left) in Aurora-4. By replacing $z_2$ of a male-speaker utterance with $z_2$ of a female-speaker utterance, an FHVAE decodes a voice-converted utterance (middle-right) that preserves the linguistic content. Audio samples are available at `https://youtu.be/VMX3IZYWYdg`.

We can observe that along each row in block 'C', the linguistic phonetic-level content, which manifests itself in the form of the spectral contour and temporal position of formants, as well as the relative position between formants, is very similar between elements; the speaker identity however changes (e.g., harmonic structure). On the other hand, for each column we see that the speaker identity remains consistent, despite the change of linguistic content. The factorization of the sequence-level attributes and the segment-level attributes of our proposed Seq2Seq-FHVAE is clearly evident. In addition, we also show examples of modifying an entire utterance in Figure 1 and 5, which achieves denoising by replacing the latent sequence variable of a noisy utterance with those of a clean utterance, and achieves voice conversion by replacing the latent sequence variable of one speaker with that of another speaker. Details of the operations we applied to modify an entire utterance as well as more larger-sized examples of different $\alpha$ values can be found in Appendix E. We also show extra latent space traversal experiments in Appendix H.

## 4.2 Quantitative Evaluation of S-Vectors – Speaker Verification

To quantify the performance of our model on disentangling the utterance-level attributes from the segment-level attributes, we present experiments on a speaker verification task on the TIMIT corpus to evaluate how well the estimated $\mu_2$ encodes speaker-level information.[5] As a sanity check, we modify Eq. 5 to estimate an alternative s-vector based on latent segment variables $z_1$ as follows: $\mu_1 = \sum_{n=1}^{\tilde{N}} g_{\mu_{z_1}}(\tilde{x}^{(n)})/(\tilde{N} + \sigma_{z_1}^2)$. We use the i-vector method [5] as the baseline, which is the representation used in most state-of-the-art speaker verification systems. They are in a low dimensional subspace of the Gaussian mixture model (GMM) mean supervector space, where the GMM is the universal background model (UBM) that models the generative process of speech. I-vectors, $\mu_1$, and $\mu_2$ can all be extracted without supervision; when speaker labels are available during training, techniques such as linear discriminative analysis (LDA) can be applied to further improve the linear separability of the representation. For all experiments, we use the fast scoring approach in [4] that uses cosine similarity as the similarity metric and compute the equal error rate (EER). More details about the experimental settings can be found in Appendix F.

We compare different dimensions for both features as well as different $\alpha$'s in Eq.3 for training FHVAE models. The results in Table 1 show that the 16 dimensional s-vectors $\mu_2$ outperform i-vector baselines in both unsupervised (Raw) and supervised (LDA) settings for all $\alpha$'s as shown in the fourth column; the more discriminatively the FHVAE model is trained (i.e., with larger $\alpha$), the better speaker

verification results it achieves. Moreover, with the appropriately chosen dimension, a 32 dimensional $\mu_2$ reaches an even lower EER at 1.34%. On the other hand, the negative results of using $\mu_1$ also validate the success in disentangling utterance and segment level attributes.

Table 1: Comparison of speaker verification equal error rate (EER) on the TIMIT test set

| Features | Dimension | $\alpha$ | Raw | LDA (12 dim) | LDA (24 dim) |
|---|---|---|---|---|---|
| i-vector | 48 | - | 10.12% | 6.25% | 5.95% |
| | 100 | - | 9.52% | 6.10% | 5.50% |
| | 200 | - | 9.82% | 6.54% | 6.10% |
| $\mu_2$ | 16 | 0 | 5.06% | 4.02% | - |
| | 16 | $10^{-1}$ | 4.91% | 4.61% | - |
| | 16 | $10^0$ | 3.87% | 3.86% | - |
| | 16 | $10^1$ | **2.38%** | **2.08%** | - |
| | 32 | $10^1$ | **2.38%** | **2.08%** | **1.34%** |
| $\mu_1$ | 16 | $10^0$ | 22.77% | 15.62% | - |
| | 16 | $10^1$ | 27.68% | 22.17% | - |
| | 32 | $10^1$ | 22.47% | 16.82% | 17.26% |

## 4.3 Quantitative Evaluation of the Latent Segment Variables – Domain Invariant ASR

Speaker adaptation and robust speech recognition in automatic speech recognition (ASR) can often be seen as domain adaptation problems, where available labeled data is limited and hence the data distributions during training and testing are mismatched. One approach to reduce the severity of this issue is to extract speaker/channel invariant features for the tasks.

As demonstrated in Section 4.2, the s-vector contains information about domains. Here we evaluate if the latent segment variables contains domain invariant linguistic information by evaluating on an ASR task: (1) train our proposed Seq2Seq-FHVAE using FBank feature on a set that covers different domains. (2) train an LSTM acoustic model [12, 35, 42] on the set that only covers partial domains using mean and log variance of the latent segment variable $z_1$ extracted from the trained Seq2Seq-FHVAE. (3) test the ASR system on all domains. As a baseline, we also train the same ASR models but use the FBank features alone. Detailed configurations are in Appendix G.

For TIMIT we assume that male and female speakers constitute different domains, and show the results in Table 2. The first row of results shows that the ASR model trained on all domains (speakers) using FBank features as the upper bound. When trained on only male speakers, the phone error rate (PER) on female speakers increases by 16.1% for FBank features; however, for $z_1$, despite the slight degradation on male speakers, the PER on the unseen domain, which are female speakers, improves by 6.6% compared to FBank features.

Table 2: TIMIT test phone error rate of acoustic models trained on different features and sets

| Train Set and Configuration | | | Test PER by Set | | |
|---|---|---|---|---|---|
| ASR | FHVAE | Features | Male | Female | All |
| Train All | - | FBank | 20.1% | 16.7% | 19.1% |
| Train Male | - | FBank | **21.0%** | 32.8% | 25.2% |
| | Train All, $\alpha = 10$ | $z_1$ | 22.0% | **26.2%** | **23.5%** |

On Aurora-4, four domains are considered, which are clean, noisy, channel, and noisy+channel (NC for short). We train the FHVAE on the development set for two purposes: (1) the FHVAE can be considered as a general feature extractor, which can be trained on an arbitrary collection of data that does not necessarily include the data for subsequent applications. (2) the dev set of Aurora-4 contains the domain label for each utterance so it is possible to control which domain has been observed by the FHVAE. Table 3 shows the word error rate (WER) results on Aurora-4, from which we can observe that the FBank representation suffers from severe domain mismatch problems; specifically, the WER

increases by 53.3% when noise is presented in mismatched microphone recordings (NC). In contrast, when the FHVAE is trained on data from all domains, using the latent segment variables as features reduce WER from 16% to 35% compare to baseline on mismatched domains, with less than 2% WER degradation on the matched domain. In addition, $\beta$-VAEs [13] are trained on the same data as the FHVAE to serve as the baseline feature extractor, from which we extract the latent variables $z$ as the ASR feature and show the result in the third to the sixth rows. The $\beta$-VAE features outperform FBank in all mismatched domains, but are inferior to the latent segment variable $z_1$ from the FHVAE in those domains. The results demonstrate the importance of learning not only disentangled, but also interpretable representations, which can be achieved by our proposed FHVAE models. As a sanity check, we replace $z_1$ with $z_2$, the latent sequence variable and train an ASR, which results in terrible WER performance as shown in the eighth row as expected.

Finally, we train another FHVAE on all domains excluding the combinatory NC domain, and shows the results in the last row in Table 3. It can be observed that the latent segment variable still outperforms the baseline feature with 30% lower WER on noise and channel combined data, even though the FHAVE has only seen noise and channel variation independently.

Table 3: Aurora-4 test word error rate of acoustic models trained on different features and sets

| Train Set and Configuration | | | Test WER by Set | | | | |
|---|---|---|---|---|---|---|---|
| ASR | {FH-,$\beta$-}VAE | Features | Clean | Noisy | Channel | NC | All |
| Train All | - | FBank | 3.60% | 7.06% | 8.24% | 18.49% | 11.80% |
| Train Clean | - | FBank | **3.47%** | 50.97% | 36.99% | 71.80% | 55.51% |
| | Dev, $\beta = 1$ | $z$ ($\beta$-VAE) | 4.95% | 23.54% | 31.12% | 46.21% | 32.47% |
| | Dev, $\beta = 2$ | $z$ ($\beta$-VAE) | 3.57% | 27.24% | 30.56% | 48.17% | 34.75% |
| | Dev, $\beta = 4$ | $z$ ($\beta$-VAE) | 3.89% | 24.40% | 29.80% | 47.87% | 33.38% |
| | Dev, $\beta = 8$ | $z$ ($\beta$-VAE) | 5.32% | 34.84% | 36.13% | 58.02% | 42.76% |
| | Dev, $\alpha = 10$ | $z_1$ (FHVAE) | 5.01% | **16.42%** | **20.29%** | **36.33%** | **24.41%** |
| | Dev, $\alpha = 10$ | $z_2$ (FHVAE) | 41.08% | 68.73% | 61.89% | 86.36% | 72.53% |
| | Dev\NC, $\alpha = 10$ | $z_1$ (FHVAE) | 5.25% | 16.52% | 19.30% | 40.59% | 26.23% |

## 5 Related Work

A number of prior publications have extended VAEs to model structured data by altering the underlying graphical model to dynamic Bayesian networks, such as SRNN [3] and VRNN [9], or to hierarchical models, such as neural statistician [7] and SVAE [18]. These models have shown success in quantitatively increasing the log-likelihood, or qualitatively generating reasonable structured data by sampling. However, it remains unclear whether independent attributes are disentangled in the latent space. Moreover, the learned latent variables in these models are not interpretable without manually inspecting or using labeled data. In contrast, our work presents a VAE framework that addresses both problems by explicitly modeling the difference in the rate of temporal variation of the attributes that operate at different scales.

Our work is also related to $\beta$-VAE [13] with respect to unsupervised learning of disentangled representations with VAEs. The boosted KL-divergence penalty imposed in $\beta$-VAE training encourages disentanglement of independent attributes, but does not provide interpretability without supervision. We demonstrate in our domain invariant ASR experiments that learning interpretable representations is important for such applications, and can be achieved by our FHVAE model. In addition, the idea of boosting KL-divergence regularization is complimentary to our model, which can be potentially integrated for better disentanglement.

## 6 Conclusions and Future Work

We introduce the factorized hierarchical variational autoencoder, which learns disentangled and interpretable representations for sequence-level and segment-level attributes without any supervision. We verify the disentangling ability both qualitatively and quantitatively on two speech corpora. For future work, we plan to (1) extend to more levels of hierarchy, (2) investigate adversarial training for disentanglement, and (3) apply the model to other types of sequential data, such as text and videos.

## Footnotes

[1] `https://github.com/wnhsu/FactorizedHierarchicalVAE`

[2]The index starts from 2 because we do not introduce the hierarchy to $\boldsymbol{z}_1$.

[3]Best viewed in color.

[4]The harmonics corresponds to horizontal dark stripes in the figure; the more widely these stripes are spaced vertically, the higher the fundamental frequency of the speaker is.

[5]TIMIT is not a standard corpus for speaker verification, but it is a good corpus to show the utterance-level attribute we learned via this task, because the main attribute that is consistent within an utterance is speaker identity, while in Aurora-4 both speaker identity and the background noise are consistent within an utterance.

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
