[Supplementary Material]

# Unsupervised Learning of Disentangled and Interpretable Representations from Sequential Data – Supplementary Materials

**Wei-Ning Hsu, Yu Zhang, and James Glass**
Computer Science and Artificial Intelligence Laboratory
Massachusetts Institute of Technology
Cambridge, MA 02139, USA
{wnhsu,yzhang87,glass}@csail.mit.edu

## A. Derivation of Sequence Variational Lower Bound

The variational lower bound for the marginal probability of a sequence $\boldsymbol{X}$ can be derived as follows:

$$
\begin{aligned}
\log p_\theta(\boldsymbol{X}) \geq & \mathcal{L}(\theta, \phi; \boldsymbol{X}) \\
= & \mathbb{E}_{q_\phi(\boldsymbol{Z}_1, \boldsymbol{Z}_2, \boldsymbol{\mu}_2 | \boldsymbol{X})} \Big[ \log \frac{p_\theta(\boldsymbol{\mu}_2) \prod_{n=1}^N p_\theta(\boldsymbol{x}^{(n)} | \boldsymbol{z}_1^{(n)}, \boldsymbol{z}_2^{(n)}) p_\theta(\boldsymbol{z}_1^{(n)}) p_\theta(\boldsymbol{z}_2^{(n)} | \boldsymbol{\mu}_2)}{q_\phi(\boldsymbol{\mu}_2) \prod_{n=1}^N q_\phi(\boldsymbol{z}_1^{(n)} | \boldsymbol{x}^{(n)}, \boldsymbol{z}_2^{(n)}) q_\phi(\boldsymbol{z}_2^{(n)} | \boldsymbol{x}^{(n)})} \Big] \\
= & \sum_{n=1}^N \mathbb{E}_{q_\phi(\boldsymbol{z}_1^{(n)}, \boldsymbol{z}_2^{(n)} | \boldsymbol{x}^{(n)})} \big[ \log p_\theta(\boldsymbol{x}^{(n)} | \boldsymbol{z}_1^{(n)}, \boldsymbol{z}_2^{(n)}) \big] \\
& - \sum_{n=1}^N \mathbb{E}_{q_\phi(\boldsymbol{z}_2^{(n)} | \boldsymbol{x}^{(n)})} \big[ D_{KL}(q_\phi(\boldsymbol{z}_1^{(n)} | \boldsymbol{x}^{(n)}, \boldsymbol{z}_2^{(n)}) || p_\theta(\boldsymbol{z}_1^{(n)})) \big] \\
& - \sum_{n=1}^N \mathbb{E}_{q_\phi(\boldsymbol{\mu}_2)} \big[ D_{KL}(q_\phi(\boldsymbol{z}_2^{(n)} | \boldsymbol{x}^{(n)}) || p_\theta(\boldsymbol{z}_2^{(n)} | \boldsymbol{\mu}_2)) \big] \quad (1) \\
& - D_{KL}(q_\phi(\boldsymbol{\mu}_2) || p_\theta(\boldsymbol{\mu}_2)). \quad (2)
\end{aligned}
$$

The expected KL-divergence in Eq. 1 of two Gaussian distributions, $q_\phi(\boldsymbol{z}_2^{(n)} | \boldsymbol{x}^{(n)})$ and $p_\theta(\boldsymbol{z}_2^{(n)} | \boldsymbol{\mu}_2)$, over a Gaussian $q_\phi(\boldsymbol{\mu}_2) = \mathcal{N}(\boldsymbol{\mu}_2 | \tilde{\boldsymbol{\mu}}_2, \sigma_{\tilde{\boldsymbol{\mu}}_2}^2 \boldsymbol{I})$ can be computed analytically. Let $J$ be the dimensionality of $\boldsymbol{z}_2$. Let $\hat{\boldsymbol{\mu}}_{\boldsymbol{z}_2}$ and $\hat{\boldsymbol{\sigma}}_{\boldsymbol{z}_2}$ denote the variational mean and standard deviation evaluated at $\boldsymbol{x}^{(n)}$,

and let $\mu_{2,j}$, $\tilde{\mu}_{2,j}$, $\hat{\mu}_{\boldsymbol{z}_2,j}$ and $\hat{\sigma}_{\boldsymbol{z}_2,j}$ denote the $j$-th element of these vectors. We have:

$$\mathbb{E}_{q_\phi(\boldsymbol{\mu}_2)}\big[D_{KL}(q_\phi(\boldsymbol{z}_2^{(n)}|\boldsymbol{x}^{(n)})||p_\theta(\boldsymbol{z}_2^{(n)}|\boldsymbol{\mu}_2))\big]$$

$$= \mathbb{E}_{q_\phi(\boldsymbol{\mu}_2)}\big[D_{KL}(\mathcal{N}(\hat{\boldsymbol{\mu}}_{\boldsymbol{z}_2},\hat{\boldsymbol{\sigma}}_{\boldsymbol{z}_2}^2)||\mathcal{N}(\boldsymbol{\mu}_2,\sigma_{\boldsymbol{z}_2}^2\boldsymbol{I}))\big]$$

$$= \mathbb{E}_{q_\phi(\boldsymbol{\mu}_2)}\Big[-\frac{1}{2}\sum_{j=1}^{J}(1+\log\frac{\hat{\sigma}_{\boldsymbol{z}_2,j}^2}{\sigma_{\boldsymbol{z}_2}^2}-\frac{(\hat{\mu}_{\boldsymbol{z}_2,j}-\mu_{2,j})^2+\hat{\sigma}_{\boldsymbol{z}_2,j}^2}{\sigma_{\boldsymbol{z}_2}^2})\Big]$$

$$= -\frac{1}{2}\sum_{j=1}^{J}(1+\log\frac{\hat{\sigma}_{\boldsymbol{z}_2,j}^2}{\sigma_{\boldsymbol{z}_2}^2}-\frac{\hat{\sigma}_{\boldsymbol{z}_2,j}^2}{\sigma_{\boldsymbol{z}_2}^2}-\mathbb{E}_{q_\phi(\boldsymbol{\mu}_2)}\big[\frac{(\hat{\mu}_{\boldsymbol{z}_2,j}-\mu_{2,j})^2}{\sigma_{\boldsymbol{z}_2}^2}\big])$$

$$= D_{KL}(\mathcal{N}(\hat{\boldsymbol{\mu}}_{\boldsymbol{z}_2},\hat{\boldsymbol{\sigma}}_{\boldsymbol{z}_2}^2)||\mathcal{N}(\tilde{\boldsymbol{\mu}}_2,\sigma_{\boldsymbol{z}_2}^2)) + \frac{J}{2}\frac{\sigma_{\tilde{\boldsymbol{\mu}}_2}^2}{\sigma_{\boldsymbol{z}_2}^2}$$

$$= D_{KL}(q_\phi(\boldsymbol{z}_2^{(n)}|\boldsymbol{x}^{(n)})||p_\theta(\boldsymbol{z}_2^{(n)}|\tilde{\boldsymbol{\mu}}_2)) + \frac{J}{2}\frac{\sigma_{\tilde{\boldsymbol{\mu}}_2}^2}{\sigma_{\boldsymbol{z}_2}^2} \tag{3}$$

The KL-divergence in Eq. 2 can also be computed analytically and rewritten as follows:

$$D_{KL}(q_\phi(\boldsymbol{\mu}_2)||p_\theta(\boldsymbol{\mu}_2))$$

$$= D_{KL}(\mathcal{N}(\tilde{\boldsymbol{\mu}}_2,\sigma_{\tilde{\boldsymbol{\mu}}_2}^2\boldsymbol{I})||\mathcal{N}(0,\sigma_{\boldsymbol{\mu}_2}^2\boldsymbol{I}))$$

$$= -\frac{1}{2}\sum_{j=1}^{J}(1+\log\frac{\sigma_{\tilde{\boldsymbol{\mu}}_2}^2}{\sigma_{\boldsymbol{\mu}_2}^2}-\frac{(\tilde{\mu}_{2,j}-0)^2+\sigma_{\tilde{\boldsymbol{\mu}}_2}^2}{\sigma_{\boldsymbol{\mu}_2}^2})$$

$$= -\frac{1}{2}\sum_{j=1}^{J}(1+\log\sigma_{\tilde{\boldsymbol{\mu}}_2}^2)-\frac{1}{2}\log 2\pi-\log p_\theta(\tilde{\boldsymbol{\mu}}_2) \tag{4}$$

By replacing Eq. 1 and 2 with Eq. 3 and 4 respectively, we rewrite the variational lower bound for a sequence $\boldsymbol{X}$ as follows:

$$\mathcal{L}(\theta,\phi;\boldsymbol{X}) = \sum_{n=1}^{N}\mathbb{E}_{q_\phi(\boldsymbol{z}_1^{(n)},\boldsymbol{z}_2^{(n)}|\boldsymbol{x}^{(n)})}\big[\log p_\theta(\boldsymbol{x}^{(n)}|\boldsymbol{z}_1^{(n)},\boldsymbol{z}_2^{(n)})\big]$$

$$-\sum_{n=1}^{N}\mathbb{E}_{q_\phi(\boldsymbol{z}_2^{(n)}|\boldsymbol{x}^{(n)})}\big[D_{KL}(q_\phi(\boldsymbol{z}_1^{(n)}|\boldsymbol{x}^{(n)},\boldsymbol{z}_2^{(n)})||p_\theta(\boldsymbol{z}_1^{(n)}))\big]$$

$$-\sum_{n=1}^{N}D_{KL}(q_\phi(\boldsymbol{z}_2^{(n)}|\boldsymbol{x}^{(n)})||p_\theta(\boldsymbol{z}_2^{(n)}|\tilde{\boldsymbol{\mu}}_2))-\frac{J}{2}\frac{\sigma_{\tilde{\boldsymbol{\mu}}_2}^2}{\sigma_{\boldsymbol{z}_2}^2}$$

$$+\frac{1}{2}\sum_{j=1}^{J}(1+\log\sigma_{\tilde{\boldsymbol{\mu}}_2}^2)+\frac{1}{2}\log 2\pi+\log p_\theta(\tilde{\boldsymbol{\mu}}_2)$$

$$= \sum_{n=1}^{N}(\mathcal{L}(\theta,\phi;\boldsymbol{x}^{(n)}|\tilde{\boldsymbol{\mu}}_2)-\frac{J}{2}\frac{\sigma_{\tilde{\boldsymbol{\mu}}_2}^2}{\sigma_{\boldsymbol{z}_2}^2})+\frac{1}{2}\sum_{j=1}^{J}(1+\log\sigma_{\tilde{\boldsymbol{\mu}}_2}^2)+\frac{1}{2}\log 2\pi+\log p_\theta(\tilde{\boldsymbol{\mu}}_2)$$

$$= \sum_{n=1}^{N}\mathcal{L}(\theta,\phi;\boldsymbol{x}^{(n)}|\tilde{\boldsymbol{\mu}}_2)+\log p_\theta(\tilde{\boldsymbol{\mu}}_2)+const,$$

where the conditional segment variational lower bound, $\mathcal{L}(\theta,\phi;\boldsymbol{x}^{(n)}|\tilde{\boldsymbol{\mu}}_2)$, is defined as:

$$\mathcal{L}(\theta,\phi;\boldsymbol{x}^{(n)}|\tilde{\boldsymbol{\mu}}_2) = \mathbb{E}_{q_\phi(\boldsymbol{z}_1^{(n)},\boldsymbol{z}_2^{(n)}|\boldsymbol{x}^{(n)})}\big[\log p_\theta(\boldsymbol{x}^{(n)}|\boldsymbol{z}_1^{(n)},\boldsymbol{z}_2^{(n)})\big]$$

$$-\mathbb{E}_{q_\phi(\boldsymbol{z}_2^{(n)}|\boldsymbol{x}^{(n)})}\big[D_{KL}(q_\phi(\boldsymbol{z}_1^{(n)}|\boldsymbol{x}^{(n)},\boldsymbol{z}_2^{(n)})||p_\theta(\boldsymbol{z}_1^{(n)}))\big]$$

$$-D_{KL}(q_\phi(\boldsymbol{z}_2^{(n)}|\boldsymbol{x}^{(n)})||p_\theta(\boldsymbol{z}_2^{(n)}|\tilde{\boldsymbol{\mu}}_2)).$$

## B. Derivation of the Inferred S-Vector

As described in Section 2.2, inference of the s-vector $\boldsymbol{\mu}_2$ of an unseen utterance $\tilde{\boldsymbol{X}} = \{\tilde{\boldsymbol{x}}^{(n)}\}_{n=1}^{\tilde{N}}$ is cast as an approximated maximum a posterior estimation problem, which uses the conditional segment variational lower bound, $\mathcal{L}(\theta, \phi; \tilde{\boldsymbol{x}}^{(n)}|\boldsymbol{\mu}_2)$, to approximate the conditional likelihood of a segment, $\log p_\theta(\tilde{\boldsymbol{x}}^{(n)}|\boldsymbol{\mu}_2)$. Let $J$ be the dimensionality of $\boldsymbol{z}_2$. Let $\hat{\boldsymbol{\mu}}_{\boldsymbol{z}_2}^{(n)}$ denote the variational mean of $\boldsymbol{z}_2$ evaluated at $\boldsymbol{x}^{(n)}$, and let $\mu_{2,j}$ and $\hat{\mu}_{\boldsymbol{z}_2,j}^{(n)}$ denote the $j$-th element of these vectors. The optimal $\boldsymbol{\mu}_2^*$ can be derived as follows:

$$
\begin{aligned}
\boldsymbol{\mu}_2^* &= \operatorname*{argmax}_{\boldsymbol{\mu}_2} \sum_{n=1}^{\tilde{N}} \mathcal{L}(\theta, \phi; \tilde{\boldsymbol{x}}^{(n)}|\boldsymbol{\mu}_2) + \log p_\theta(\boldsymbol{\mu}_2) \\
&= \operatorname*{argmax}_{\boldsymbol{\mu}_2} \sum_{n=1}^{\tilde{N}} -D_{KL}(q_\phi(\boldsymbol{z}_2^{(n)}|\tilde{x}^{(n)})||p_\theta(\boldsymbol{z}_2^{(n)}|\boldsymbol{\mu}_2)) + \log p_\theta(\boldsymbol{\mu}_2) \\
&= \operatorname*{argmax}_{\boldsymbol{\mu}_2} \sum_{n=1}^{\tilde{N}} \sum_{j=1}^{J} \frac{-(\hat{\mu}_{\boldsymbol{z}_2,j}^{(n)} - \mu_{2,j})^2}{\sigma_{\boldsymbol{z}_2}^2} + \sum_{j=1}^{J} \frac{-(\mu_{2,j} - 0)^2}{\sigma_{\boldsymbol{\mu}_2}^2} \\
&= \operatorname*{argmax}_{\boldsymbol{\mu}_2} f(\boldsymbol{\mu}_2),
\end{aligned}
$$

where $f(\cdot)$ is a concave quadratic function that has only one maximum point. We then have:

$$
\left.\frac{\partial f(\boldsymbol{\mu}_2)}{\partial \boldsymbol{\mu}_2}\right|_{\boldsymbol{\mu}_2 = \tilde{\boldsymbol{\mu}}_2^*} = 0
$$

$$
\boldsymbol{\mu}_2^* = \frac{\sum_{n=1}^{\tilde{N}} \hat{\boldsymbol{\mu}}_{\boldsymbol{z}_2}^{(n)}}{\tilde{N} + \sigma_{\boldsymbol{z}_2}^2/\sigma_{\boldsymbol{\mu}_2}^2}
$$

## C. FHVAE Model and Training Configurations

For the Seq2Seq-FHVAE model, each $LSTM$ network consists of one layer with 256 hidden units, while each $MLP$ network is one layer with the output dimension equal to the variable whose mean or log variance the $MLP$ parameterizes, and variances $\sigma_{\boldsymbol{z}_1}^2 = \sigma_{\boldsymbol{\mu}_2}^2 = 1$, $\sigma_{\boldsymbol{z}_2}^2 = 0.25$. We experiment with various dimensions for the latent variable $\boldsymbol{z}_1$ and $\boldsymbol{z}_2$. All models were trained with stochastic gradient descent using a mini-batch size of 256 to minimize the negative discriminative segment variational lower bound plus an $L2$-regularization with weight $10^{-4}$. The Adam [4] optimizer is used with $\beta_1 = 0.95$, $\beta_2 = 0.999$, $\epsilon = 10^{-8}$, and initial learning rate of $10^{-3}$. Training continues for 500 epochs unless the segment variational lower bound on the development set does not improve for 50 epochs. The $\boldsymbol{\mu}_2$ for the sequences in the development set and the test set is estimated using the closed form solution in Section 2.2.

## D. Comparison of Seq2Seq-FHVAE and Alternative Architectures

Here we study the performance of our proposed architecture by replacing the LSTM module with three baseline architectures: a fully-connected feed-forward network (FC), a vanilla recurrent neural network (RNN), and a gated recurrent neural network (GRU) [1]. All the models have one hidden layer with 16 dimensions for both $\boldsymbol{z}_1$ and $\boldsymbol{z}_2$, and are trained with $\alpha = 0$. For the FC model, the entire segment is flattened and feed to the fully-connected layers; therefore the temporal structure is simply ignored.

Table 1 shows the segment variational lower bound on the TIMIT test set. We can see that the recurrent models (RNN, GRU, LSTM) outperform the feed-forward model using fewer parameters, which demonstrates the importance of considering the temporal structure within a segment. Figure 1 shows the reconstruction results using the FC model and the LSTM model. The LSTM model reconstructs sharper images that preserves more speech detail, and, in particular, presents superior high frequency harmonic structure that does the FC model, as highlighted in the red boxes.

Table 1: TIMIT test set segment variational lower bound results on different model architectures.

| Models | #Hidden Units | #Params | $\mathcal{L}(\theta, \phi; \boldsymbol{x}^{(n)})$ |
|--------|---------------|---------|------------------|
| FC | 512 | 3.3M | -348.63 |
| RNN | 256 | 0.3M | -261.19 |
| GRU | 256 | 0.8M | -158.42 |
| LSTM | 256 | 1.1M | **-143.80** |

Figure 1: Three examples from different speakers. Within each example, from left to right are 1) the original segment, 2) FC reconstructed segment, and 3) LSTM reconstructed segment. The leftmost images show expanded views of the higher frequency harmonic structure (horizontal dark bands) of the spectrogram suggesting that the LSTM reconstruction is superior to the FC model.

## E. Transformation of Speaker and Noise Conditions

Figure 2 shows the zoomed-in version of the left part in Figure 4, from which we can observe the harmonic patterns more clearly. In Figure 3, we illustrate the results of the same experiments, but use the model trained on the Aurora-4 corpus instead. In particular, we sample two speakers, 441 and 443, from the test set and choose four noise conditions: clean, car, babble, and restaurant, without the microphone channel effect. Furthermore, since the noise is artificially added to each clean utterance in the test set, we can actually choose the corresponding segment in different noise conditions for a given speaker. Same eight examples are used in both block 'A' and block 'B', which results in 64 combinations of latent segment variables and latent sequence variables in total. It can be observed that the latent sequence variables capture not only the speaker information, but also the noise information, which are both sequence-level attributes. Therefore, when modifying the latent sequence variables, we can not only transform speaker identities, but also carry out denoising or noise corruption. Moreover, the disentanglement is evident for both the model trained without discriminative training ($\alpha = 0$) and the model trained with discriminative training ($\alpha = 10$).

In addition to transforming a single segment, one may also be interested in transforming a target sequence $\boldsymbol{X}_{tar}$ to be of a different speaker or a different noise condition of a reference sequence $\boldsymbol{X}_{ref}$. Mathematically, it means mapping the distribution of the latent sequence variable from that of $\boldsymbol{X}_{tar}$ to that of $\boldsymbol{X}_{ref}$. Since the distributions are both Gaussian with the same covariance matrices, centered at their own s-vectors, $\boldsymbol{\mu}_{2,tar}$ and $\boldsymbol{\mu}_{2,ref}$, a simple solution is to shift the latent sequence variable by the s-vector difference $\Delta\boldsymbol{\mu}_2 = \boldsymbol{\mu}_{2,ref} - \boldsymbol{\mu}_{2,tar}$. Therefore, we transform a target utterance given a reference utterance by shifting the $\boldsymbol{z}_2$ of each segment from the target utterance by $\Delta\boldsymbol{\mu}_2$, and then decode-and-concatenate each segment using the unmodified $\boldsymbol{z}_1$ and the modified $\boldsymbol{z}_2$. Figure 4, 5, 6, and 7 shows examples of modifying entire utterances, which achieves voice conversion and denoising respectively.

## F. More Details about the Speaker Verification Experiments

Verification performance is reported in terms of equal error rate (EER), where the false rejection rate equals the false acceptance rate. For our baseline system, we use the i-vectors [2] provided by Kaldi [5], which are extracted using Mel-frequency cepstral coefficients (MFCCs), plus delta and delta-delta after voice activity detection (VAD). A full-covariance gender-independent UBM with 2048 mixtures was trained on the training set and the i-vector dimensionality is tuned on the development set. The verification pairs were created from the test set as target/non-target. There are in total 24 speakers and 18,336 pairs for testing. For all the Seq2Seq-FHVAE model, $\boldsymbol{z}_1$ and $\boldsymbol{z}_2$ have

Figure 2: Examples generated by varying different latent variables of a FHVAE model trained with $\alpha = 10$ on TIMIT dataset. The green block 'A' contains four reconstructed examples. The red block 'B' contains ten original examples on the first row and the corresponding reconstructed examples on the second row. The entry on the $i$-th row and the $j$-th column in the blue block 'C' is the reconstructed example using the latent segment variable $z_1$ of the $i$-th row from the block 'A' and the latent sequence variable $z_2$ of the $j$-th column from the block 'B.'

the same dimension, and we use the closed form solution of the inferred s-vector as mentioned in Section 2.2 to represent each utterance for verification.

## G. More Details about the Domain Invariant ASR Experiments

The Gaussian mixture model-hidden Markov models (GMM-HMM) systems are built first to generate the senone (tied triphone HMM state) alignments for the later neural network acoustic model training, which replaces the GMM acoustic model. In both tasks (TIMIT and Aurora-4), the GMM-HMM system is built with Kaldi [5] using standard recipes. We use the LSTM [3] for the acoustic model in our hybrid DNN-HMM system, which are implemented using the CNTK [7] toolkit. Our training recipe follows [8]. The baseline uses 80-dimensional FBank features as input. The model has 3 LSTM-projection layers [6], where each layer has 1024 cells and the output is projected to a 512 dimensional space. The truncated BPTT is used to train the LSTM that unrolls 20 frames; 40 utterances are processed in parallel to form a mini-batch. For the Seq2Seq-FHVAE model, we use the same configuration as the one that achieved the best result on the speaker verification task: both $z_1$ and $z_2$ are 32 dimensional, and the weight $\alpha = 10$ for discriminative training. For the VAE model, the dimension of the latent variable $z$ is 64, and the number of hidden units of the LSTM encoder is 512. We doubled both the latent variable dimension and the number of hidden units for the encoder compared to the FHVAE model because the VAE model only has one set of latent variables and one encoder. Therefore, both the FHVAE and VAE models would have a comparable number of parameters as well as latent space dimensionalities.

Figure 3: Examples generated by varying $z_1$ and $z_2$ of two FHVAE models trained with $\alpha = 0$ and $\alpha = 10$ respectively on Aurora-4 dataset. The green block 'A' and the red block 'B' contains the same eight examples from the test set. In block 'B,' original examples are shown on the first row and the corresponding reconstructed examples are shown on the second row. The entry on the $i$-th row and the $j$-th column in the blue block 'C' is the reconstructed example using the latent segment variable $z_1$ of the $i$-th row from the block 'A' and the latent sequence variable $z_2$ of the $j$-th column from the block 'B.'

## H. FHVAE Latent Space Traversal

In this section, we present a qualitative analysis of traversing a single latent sequence variable or latent segment variable over the range $[-3, 3]$, while keeping the remaining latent variables fixed. Each row corresponds to a different seed $(z_1, z_2)$ pair, inferred from some seed segment randomly drawn from the test set. The leftmost column in each figure shows the seed segments for each row. We use the same five seed segments for traversing each latent variable. The FHVAE model is trained on TIMIT with $\alpha = 0$, and a 200 dimensional log-magnitude spectrum is used for frame feature representations.

Figures 8 and 9 show examples of traversing four different latent segment variables, $z_1$, while keeping the latent sequence variables fixed. It can be observed that these latent segment variables encode the information of segment-level attributes in speech data, such as rising/falling F2, back vowel/front vowel, vowel/fricative, and closure/non-closure.

In contrast, Figures 10 and 11 illustrate examples for traversing four different latent sequence variables, $z_2$, while keeping the latent segment variables fixed. It can be seen the spectral contour, temporal position, and relative frequency-axis position of formants remain almost intact when traversing these latent sequence variables. The attributes being changed when traversing these latent sequence variables are more related to sequence-level attributes, such as harmonic patterns (F0), volume, offsets of formant frequencies. The results again demonstrate the ability of our proposed FHVAE to not

Figure 4: FHVAE ($\alpha = 0$) decoding results of three combinations of *latent segment variables* $z_1$ and *latent sequence variables* $z_2$ from two utterances in Aurora-4: a clean one (top-left) and a noisy one (bottom-left). FHVAEs learn to encode local attributes, such as linguistic content, into $z_1$, and encode global attributes, such as noise level, into $z_2$. Therefore, by replacing $z_2$ of a noisy utterance with $z_2$ of a clean utterance, an FHVAE decodes a denoised utterance (middle-right) that preserves the linguistic content. Reconstruction results of the clean and noisy utterances are also shown on the right. Audio samples are available at `https://youtu.be/naJZITvCfI4`.

Figure 5: FHVAE ($\alpha = 0$) decoding results of three combinations of *latent segment variables* $z_1$ and *latent sequence variables* $z_2$ from one clean utterance (top-left) and one utterance with car noise (bottom-left) in Aurora-4. By replacing $z_2$ of a noisy utterance with $z_2$ of a clean utterance, an FHVAE decodes a denoised utterance (middle-right) that preserves the linguistic content. Audio samples are available at `https://youtu.be/pOP2DVZWRjM`.

only learn disentangled representations, but also enable interpretation of the information captured by different sets of latent variables.

Figure 6: FHVAE ($\alpha = 0$) decoding results of three combinations of *latent segment variables* $z_1$ and *latent sequence variables* $z_2$ from one male-speaker utterance (top-left) and one female-speaker utterance (bottom-left) in Aurora-4. By replacing $z_2$ of a male-speaker utterance with $z_2$ of a female-speaker utterance, an FHVAE decodes a voice-converted utterance (middle-right) that preserves the linguistic content. Audio samples are available at `https://youtu.be/VMX3IZYWYdg`.

Figure 7: FHVAE ($\alpha = 0$) decoding results of three combinations of *latent segment variables* $z_1$ and *latent sequence variables* $z_2$ from one female-speaker utterance (top-left) and one male-speaker utterance (bottom-left) in Aurora-4. By replacing $z_2$ of a female-speaker utterance with $z_2$ of a male-speaker utterance, an FHVAE decodes a voice-converted utterance (middle-right) that preserves the linguistic content. Audio samples are available at `https://youtu.be/Rurj2ByNRs8`.

# Latent Segment Variable Traversal

Figure 8: Traversing two different latent segment variables with five seed segments from the TIMIT test set using an FHVAE model trained on TIMIT with $\alpha = 0$.

# Latent Segment Variable Traversal

Figure 9: Traversing another two different latent segment variables with five seed segments from the TIMIT test set using an FHVAE model trained on TIMIT with $\alpha = 0$.

# Latent Sequence Variable Traversal

Figure 10: Traversing two different latent sequence variables with five seed segments from the TIMIT test set using an FHVAE model trained on TIMIT with $\alpha = 0$.

Figure 11: Traversing another two different latent sequence variables with five seed segments from the TIMIT test set using an FHVAE model trained on TIMIT with $\alpha = 0$.