[Reviews · NeurIPS 2017]

Reviewer 1



This paper presents a factorized hierarchical variational autoencoder applied to unsupervised sequence modeling. The main claim of the paper is that the proposed model can disentangle the sequence representation into frame-level and sequence-level components. The sequence-level representation can be used for applications such as speaker verification, without any supervision in learning the representation, and they show that it is better than competitive unsupervised baselines such as using i-vector representations. The model is a mostly straightforward adaptation of sequential VAEs, with the addition of a discriminative regularizer that encourages sequence-level features to be able to predict sequence indices. Does this mean the actual index of a sequence in the training set? Or an index into list of sequences comprising a single training example? In experiments, the authors show that the proposed method achieves strong results on a variety of discriminative tasks using the proposed unsupervised feature learning method. Questions: - “Note that the mean of the posterior distribution of μ2 is not directly inferred from X, but instead is built into the set of inference model parameters with one for each sequence.” Does this refer to training set sequences? If that is the case, does this not present a problem when applying the model to new data? In general the usage of the terms “sequence” and “frame” are somewhat confusing throughout the paper.
 - How do the frame-level features perform on speaker verification? If there is some simple heuristic that could be used to derive a sequence embedding from the frame features, that would be a very interesting control experiment. Likewise, is there a way to derive frame-level features from sequence-level components of the model? This would be a way to quantitatively assess the disentangling. (It looks like table 4 row 5 may already contain this second control experiment). - Have you experimented with adversarial losses using sequence index prediction? For example one could encourage frame-level features to not be able to predict sequence indices, and vice-versa. - Have you tried generating audio from this model, and if so how to the samples sound?

Reviewer 2



The paper proposes to the Factorized Hierarchical VAE for unsupervised hierarchical representation learning in sequential data. The authors pursue computational efficiency by making the inference possible at frame (sub-sequence) level. To complement the induced problem of this decoupling, the authors also propose to augment the objective with the discriminative objective. Qualitative and quantitative evaluations are performed by visualizing the learned latent features and on speaker verification and word error rate with i-vector baseline. The paper is well-written and easy to understand. I enjoyed reading the paper. It also tackles an important problem of learning the hierarchical representation of sequential data in an unsupervised way. It would have been nice to add another baseline which works without frame-level decoupling. Then, in addition to the accuracy, having some learning curves (for varying sequence lengths) in comparison of the baseline would be interesting because the main claim is to make the inference scalable with decoupling at the frame level. I found the comparison of different architectures in Table 1. is somewhat not much important.

Reviewer 3



Summary This paper proposes a hierarchical probabilistic generative model for sequential data. The proposed model structure distinguishes between higher-level sequence and lower-level frame latent variables in order to capture multiple scales. A discriminative regularizer is added to the ELBO to encourage diversity across sequence latent variables with the aim of disentangling the latent representations. This work is related to approaches for learning disentangled representations in probabilistic generative models, such as DC-IGN (Kulkarni et al. 2015), InfoGAN (Chen et al. 2016), and Beta-VAE (Higgins et al. 2017). It is also related to hierarchical VAE-style models such as Structured VAEs (Johnson et al. 2016) and the Neural Statistician (Edwards & Storkey 2017) and latent variable models for sequential data including VRNN (Chung et al. 2015), SRNN (Fraccaro et al. 2016) and the hierarchical multiscale RNN of Chung et al. (2017). Strengths - Disentanglement in sequential data is not studied sufficiently in current literature - Seq2Seq model within each frame is sensible - Paper is overall well-written and clear Weaknesses - The proposed hierarchical model makes frame-level independence assumptions - Discriminative objective is reminiscent of a supervised objective and thus it not clear that the resulting disentanglement of latent representations is in fact unsupervised - Experiments are somewhat incomplete, both in comparing to baselines and in empirically validating disentanglement Quality Two design decisions regarding the graphical model did not make sense to me. First, if z2 is meant to capture sequence-level information, why is it placed inside the frame-level plate? Second, modeling frames independently for a given sequence discards the correlations between frames. Even if this facilitates learning at the frame level, it does not seem to be an entirely appropriate assumption for sequential data. I am wondering if the authors can clarify the motivation behind these choices. In particular, the first decision seems to necessitate the addition of the discriminative regularizer (equation between ll. 110-111). However such an objective could be naturally incorporated into the model by adding an optional fully observed sequence-level side information variable y^{(i)} which is generated from z2. For the variational lower bound results, I would suggest adding a comparison to baseline models such as VRNN (Chung et al. 2015) or SRNN (Fraccaro et al. 2016). The disentanglement properties of the proposed FHVAE seem to be due to the discriminative objective, since the speaker verification error rate drops as alpha increases. However, since in TIMIT the sequences vary primarily on speaker identity (as mentioned in the paper), the discriminative objective is roughly equivalent to a supervised loss in which the goal is to predict speaker identity. In this case, the disentanglement is no longer unsupervised as originally claimed. Adding a speaker verification experiment on a dataset where the sequences vary less based on speaker identity would help clarify the role of the discriminative objective in disentangling the latent representation. Clarity The paper is generally well-written and structured clearly. The notation could be improved in a couple of places. In the inference model (equations between ll. 82-83), I would suggest adding a frame superscript to clarify that inference is occurring within each frame, e.g. q_{\phi}(z_2^{(n)} | x^{(n)}) and q_{\phi}(z_1^{(n)} | x^{(n)}, z_2^{(n)}). In addition, in Section 3 it was not immediately clear that a frame is defined to itself be a sub-sequence. Originality The paper is missing a related work section and also does not cite several related works, particularly regarding RNN variants with latent variables (Fraccaro et al. 2016; Chung et al. 2017), hierarchical probabilistic generative models (Johnson et al. 2016; Edwards & Storkey 2017) and disentanglement in generative models (Higgins et al. 2017). The proposed graphical model is similar to that of Edwards & Storkey (2017), though the frame-level Seq2Seq makes the proposed method sufficiently original. The study of disentanglement for sequential data is also fairly novel. Significance Unsupervised discovery of disentangled latent representations is an important problem. There has been some prior work in learning disentangled representations for image data but to my knowledge not for sequential data. However the proposed discriminative objective sufficiently resembles supervised identity prediction that it is not clear that the proposed method actually addresses the problem. Chen, Xi, et al. "Infogan: Interpretable representation learning by information maximizing generative adversarial nets." NIPS 2016. Chung, Junyoung, et al. "A recurrent latent variable model for sequential data." NIPS 2015. Chung, Junyoung, Sungjin Ahn, and Yoshua Bengio. "Hierarchical multiscale recurrent neural networks." ICLR 2017. Edwards, Harrison, and Amos Storkey. "Towards a neural statistician." ICLR 2017. Fraccaro, Marco, et al. "Sequential neural models with stochastic layers." NIPS 2016. Higgins, Irina, et al. "beta-VAE: Learning basic visual concepts with a constrained variational framework." ICLR 2017. Johnson, Matthew, et al. "Composing graphical models with neural networks for structured representations and fast inference." NIPS 2016. Kulkarni, Tejas D., et al. "Deep convolutional inverse graphics network." NIPS 2015. ----- **Edit:** I have read the authors' rebuttal. The clarification was helpful regarding the importance of the discriminative regularizer in achieving disentanglement. I would recommend emphasizing in the paper (1) the presence of multiple utterances per speaker and (2) the speaker verification results are still good when alpha = 0. I am updating my rating of this paper from 4 to 6.